# Blastocyst-like Structures in the Peripheral Retina of Young Adult Beagles

**DOI:** 10.3390/ijms25116045

**Published:** 2024-05-30

**Authors:** Tsunehiko Ikeda, Denan Jin, Shinji Takai, Kimitoshi Nakamura, Emika Nemoto, Shota Kojima, Hidehiro Oku

**Affiliations:** 1Department of Ophthalmology, Osaka Kaisei Hospital, Osaka 532-0003, Osaka, Japan; 2Department of Ophthalmology, Osaka Medical and Pharmaceutical University, Takatsuki 569-8686, Osaka, Japan; emika.nemoto@ompu.ac.jp (E.N.); shota@ompu.ac.jp (S.K.); hidehirooku@aol.com (H.O.); 3Department of Innovative Medicine, Graduate School of Medicine, Osaka Medical and Pharmaceutical University, Takatsuki 569-8686, Osaka, Japan; denan.jin@ompu.ac.jp (D.J.); shinji.takai@ompu.ac.jp (S.T.); 4Nakamura Eye Clinic, Matsumoto 390-0811, Nagano, Japan; nakamura-ganka01@tuba.ocn.ne.jp

**Keywords:** blastocyst-like structure, inner cell mass, trophectoderm, retinal pigment epithelial cell, retinal stem cell, DAPI, nestin, octamer-binding transcription factor 4 (Oct4), Nanog, sex-determining region Y-box 2 (Sox2), cytokeratin 18 (CK18), caudal type homeobox 2 (CDX2), retinal pigment epithelium-specific protein 65 kDa (RPE65), yes-associated protein 1 (YAP1), very small embryonic-like stem cells (VSELs)

## Abstract

In this immunohistological study on the peripheral retina of 3-year-old beagle dogs, excised retina specimens were immunostained with antibodies against nestin, Oct4, Nanog, Sox2, CDX2, cytokeratin 18 (CK 18), RPE65, and YAP1, as well as hematoxylin and DAPI, two nuclear stains. Our findings revealed solitary cysts of various sizes in the inner retina. Intriguingly, a mass of small round cells with scant cytoplasms was observed in the cavity of small cysts, while many disorganized cells partially occupied the cavity of the large cysts. The small cysts were strongly positive for nestin, Oct4, Nanog, Sox2, CDX2, CK18, and YAP1. RPE65-positive cells were exclusively observed in the tissue surrounding the cysts. Since RPE65 is a specific marker of retinal pigment epithelial (RPE) cells, the surrounding cells of the peripheral cysts were presumably derived from RPE cells that migrated intraretinally. In the small cysts, intense positive staining for nestin, a marker of retinal stem cells, seemed to indicate that they were derived from retinal stem cells. The morphology and positive staining for markers of blastocyst and RPE cells indicated that the small cysts may have formed structures resembling the blastocyst, possibly caused by the interaction between retinal stem cells and migrated RPE cells.

## 1. Introduction

Peripheral retinal degenerations include typical and reticular peripheral cystoid degeneration [1,2], lattice degeneration [3], and paving-stone degeneration [4]. Typical peripheral cystoid degeneration, consisting of multitudinous close-packed microcysts, is considered to be age-dependent due to its prevalence increasing with age [5,6]. However, solitary peripheral cysts have occasionally been observed in younger generations [7,8,9]. Many dogs, including beagles, show an age-related progression of myopia [10] that is frequently associated with peripheral cystoid degeneration [11,12]. To elucidate the cause of the cystic lesions in the peripheral retina, we performed a histopathological study using four eyes of two young adult beagles and found small solitary cysts in the peripheral retina resembling a blastocyst in early embryogenesis.

Blastocysts are composed of an inner cell mass (embryoblast), an outer cell layer (trophectoderm), and a fluid-filled cavity (blastocele) [13,14]. The inner cells of a blastocyst, which are characterized by large nuclei and scant cytoplasms, possess pluripotency to differentiate into all somatic cells in the body, and can be isolated and cultured to produce pluripotent embryonic stem (ES) cells [14,15]. The trophectoderm gives rise to placental trophoblasts [16], and totipotent cells, such as fertilized eggs and blastomeres (probably up to the four-cell stage), have the ability to form blastocysts that differentiate into extraembryonic tissues (i.e., placenta, yolk sac, etc.) in addition to all somatic cells [17].

The findings in our previous study showed that retinal pigment epithelium-specific protein 65 kDa (RPE65), a specific marker of retinal pigment epithelial (RPE) cells, was expressed in the peripheral area of lattice degeneration [18], probably due to the involvement of an intraretinal migration of RPE cells [19,20]. Based on those findings, in this present study we immunostained the peripheral cysts in retinas excised from young adult beagle dogs using various markers of blastocyst and RPE cells to investigate the pathogenesis of those cystic lesions.

## 2. Results

### 2.1. Fundoscopic Examination of the Peripheral Retina

RetCam imaging of the peripheral retina showed the presence of multiple grayish cystic lesions in the area adjacent to the ora serrata (Figure 1a; white arrowheads).

### 2.2. Stereomicroscopy Findings of the Peripheral Retina Specimens

Stereomicroscopy examination of the retina obtained from one of the four enucleated eyes of two dogs showed several solitary cysts that varied in size in the peripheral retina near the ora serrata (Figure 1b, white arrowheads).

### 2.3. Nestin Immunostaining

Immunostaining for nestin showed small round cysts and large oval cysts present in the inner part of the peripheral retina (Figure 2a–c). Low magnification microscopic images of the peripheral retina showed that the small cysts were scattered mostly within the area approximately 1000 μm from the ora serrata (Figure 2a). On the other hand, the large cysts were observed at a distance of more than 2000 μm from the ora serrata (Figure 2a). No neovascular-like changes were observed in the peripheral retina. High magnification microscopic images showed that the small cyst cavity was partially occupied by a mass of small round cells (i.e., a diameter of less than 5 μm) with scant cytoplasms in the fluid-filled cavity (Figure 2b), while the large cyst cavity contained a considerable number of disorganized and fragmented inner cells (Figure 2c). The wall of the small and large cysts was mostly comprised of a single layer of flattened cells.

Immunostaining revealed that both the round nuclei and the scant cytoplasms (thin rim) of the inner cells of the small cysts were nestin positive, although the staining was much stronger in the cytoplasms than in the nuclei (Figure 2b). The outer layer cells of the small cysts were also positively stained for nestin, and more intense staining was observed in the area where the inner cell cluster was attached to the cyst wall in an unusual pattern (Figure 2b). The attachment site was almost always located on the scleral side of the small cysts. The large cysts were weakly stained for nestin, although a part of the scleral side of the large cyst wall was more intensely stained for nestin (Figure 2c). The tissue surrounding the small and large cysts was nestin negative (Figure 2b,c). The retinal cell nuclei outside the surrounding tissue of the cysts were strongly stained with hematoxylin in a coarse granular pattern indicating the presence of condensed heterochromatin, whereas the inner cell nuclei of the small cysts were homogeneously stained in a light blue color, thus suggesting that the inner cell nuclei contained mostly euchromatin (Figure 2b).

### 2.4. 4′,6-Diamidino-2-phenylindole (DAPI) Fluorescence Staining

DAPI nuclear staining showed that small round cysts with a diameter of 40–50 μm and large oval cysts with a minor axis length of 50–80 μm and a major axis length of 100–200 μm were present in the inner part of the peripheral retina (Figure 3a,c). The inner cell nuclei of the small cysts exhibited a donut-like staining pattern resembling those of the oocyte nucleus at the germinal vesicle stage [21], which was not observed in the nuclei of the outer layer cells of the cysts (Figure 3b). Most of the inner cell nuclei of the large cysts also exhibited a donut-like staining pattern, although the nuclear diameter of the inner cells of the large cysts was smaller than that of the small cysts (Figure 3d).

### 2.5. Octamer-Binding Transcription Factor 4 (Oct4) and Nanog Immunostaining

Immunostaining revealed that the inner cell cytoplasms and the outer layer cells of the small cysts were strongly positive for Oct4 (Figure 4a). More intense staining for Oct4 was observed in the area where the inner cell cluster attached to the small cyst wall (Figure 4a). The attachment site was almost always located on the scleral side of the cysts. The inner cell nuclei of the small cysts were weakly positive for Oct4 (Figure 4a). The outer layer cells of the large cysts were moderately stained for Oct4, whereas the inner cells of the large cysts were Oct4 negative (Figure 4b). The inner cells of the large cysts were loosely dispersed, whereas those of the small cysts were tightly compacted. The inner cell cytoplasms and the outer layer cells of the small cysts were intensely positive for Nanog, while the inner cell nuclei of the small cysts were weakly positive for Nanog (Figure 4c). The large cysts were less intensely stained for Nanog than the small cysts (Figure 4c,d). The tissue surrounding the small and large cysts were moderately stained for Oct4, while they were intensely stained for Nanog (Figure 4a–d).

### 2.6. Sex-Determining Region Y-Box 2 (Sox2) and Caudal Type Homeobox 2 (CDX2) Immunostaining

Immunostaining revealed that the inner cell cytoplasms and the outer layer cells of the small cysts were moderately positive for Sox2 (Figure 5a). Intense staining for Sox2 was observed in the area where the inner cell cluster attached to the small cyst wall on the scleral side (Figure 5a). The inner cell nuclei of the small cysts were partially and weakly stained for Sox2 and were homogenously stained with hematoxylin in a light color, whereas the retinal cell nuclei outside of the surrounding tissue of the cysts were stained with hematoxylin in a coarse granular pattern (Figure 5a). Notably, the inner cell nuclei of the small cysts were extremely smaller than those of normal retinal cells. The fibroblast-like cells in the surrounding tissue of the small cysts were moderately stained for Sox2. The inner cell cytoplasms of the large cysts were moderately positive for Sox2 (Figure 5b). The inner cell nuclei and the outer layer cells of the large cysts were weakly stained for Sox2 (Figure 5b).

Immunostaining revealed that the inner cell nuclei and the outer layer cells of the small cysts were intensely positive for CDX2, a marker of trophectoderm (Figure 5c) [22]. The peripheral region of the inner cell nuclei of the small cysts was also intensely stained for CDX2 (Figure 5c). The inner cell cytoplasms and the outer layer cells of the large cysts were moderately positive for CDX2, while the inner cell nuclei of the large cysts were faintly positive (Figure 5d). The surrounding tissue of the small and large cysts was completely negative for CDX2. Like with the Oct4 staining (Figure 4a,c), SOX2 and CDX2 staining showed that the inner cells of the small cysts were tightly compacted, whereas those of the large cysts were loosely dispersed and partially fragmented (Figure 5a,d).

### 2.7. Cytokeratin 18 (CK18) and RPE65 Immunostaining

Immunostaining revealed that the thin rim of the inner cell cytoplasms and the outer layer cells of the small cysts were strongly positive for CK18 (Figure 6a). Intense staining for CK18 was observed in the area where the inner cell mass attached to the cyst wall on the scleral side (Figure 6a), while the inner cell nuclei of the small cysts were lightly stained for CK18 (Figure 6a). The staining pattern of CK18 in the small cysts was similar to that of nestin and Sox2. The outer layer cells of the large cysts on the scleral side were cuboidal-shaped and were densely stained for CK18 (Figure 6b). Other parts of the outer layer cells and the inner cells of the large cysts were lightly stained for CK18 (Figure 6b). No obvious staining of CK18 was observed in the surrounding tissue of the small and large cysts (Figure 6a,b), resembling the staining pattern of nestin and CDX2 (Figure 2a and Figure 5a).

Intense staining for RPE65 was observed on the vitreal side of the surrounding tissue of the small and large cysts (Figure 6c,d), while other parts of the surrounding tissue were moderately stained. The inner cells and the outer layer cells of the small and large cysts were scarcely stained for RPE65 (Figure 6c,d).

The inner cell nuclei of the small cysts stained for CK18 and RPE65 showed a peculiar hematoxylin staining pattern where multiple clustered or dispersed rod-like structures were intensely stained (Figure 6a,c). Morphology and hematoxylin stainability seemed to indicate that these intranuclear structures were clusters of chromosomes mostly synchronized in metaphase. A normal monkey retina was used as a negative control for CK18 and RPE65 staining. RPE65 is reportedly expressed in red/green cones [23], however, as with CK18, no apparent staining for RPE65 was observed in the monkey neural retina (Figure 7a,b). RPE65 was also negatively stained in the photoreceptor layer of the beagle retina.

### 2.8. Yes-Associated Protein 1 (YAP1) Immunostaining

The vitreal side of the outer layer cells of the small cysts was intensely stained for YAP1 in both nuclei and cytoplasms, whereas their inner cells were scarcely stained (Figure 8a). The vitreal side of the outer layer cells of the large cysts was more intensely stained for YAP1 (Figure 8b). No obvious staining for YAP1 was observed in the inner cells of the large cysts (Figure 8b). Intense staining for YAP1 was observed in the surrounding tissue on the vitreal side of the small and large cysts (Figure 8a,b). Other parts of the tissue surrounding the cysts were weakly or moderately stained (Figure 8a,b). The YAP1 staining pattern of the surrounding tissue was similar to that of RPE65.

### 2.9. Von Willebrand Factor (vWF) Immunostaining

The outer layer of the small and large cysts was intensely stained with vWF. A tiny lumen observed at the upper right of the small cyst did not contain inner cells (Figure 8a). The wall of the lumen was also stained with vWF, probably indicating that this tissue was a blood vessel.

The immunostaining pattern for each marker in the small and large cysts are summarized in Table 1.

## 3. Discussion

As mentioned above, peripheral cystoid degeneration can be classified as either typical cystoid degeneration that appears as a myriad of close-packed tiny microcysts in the outer plexiform/inner nuclear layer that progresses with age [1,2], or reticular cystoid degeneration that appears in the nerve fiber layer as similar microcysts, which can occur at any age [2]. Even though peripheral cystoid degeneration is most commonly observed in an aged retina [1], the pathogenesis of the disease is poorly understood. Due to the previously described high prevalence of peripheral cystoid degeneration in dogs [11,12], we performed this immunohistological study of the peripheral retina of young adult beagles to clarify the pathogenesis of the peripheral cystic lesions. Contrary to our expectations and the findings in the previous literature [11,12], stereomicroscopy examination of the canine peripheral retina revealed multiple scattered solitary cysts that varied in size instead of common peripheral cystoid degeneration with numerous closely packed microcysts.

Reportedly, solitary cysts, i.e., cysts that are solely composed of a wall and inner fluid, are occasionally observed in the peripheral retina [7,8,9]. In a study by Pei et al. [7], the authors reported that in a 7-year-old human eye, both small and medium size solitary cysts were observed within the neural retina very close to the ora serrata. Moreover, Johnsen et al. [8] reported that a robust expression of nestin, a neural/progenitor stem cell marker [24], was observed around the solitary cysts in the peripheral retina of a healthy human eye.

Based on the findings in that latter report [8], in this present study involving young adult beagle dogs, we first immunostained the sections of the peripheral cystic lesions of the retina for nestin, and found a mass of inner cells present in the cavity of solitary cysts, which is contrary to the findings in those previous reports [7,8,9]. In addition, our findings revealed two types of solitary cysts located in the inner retinal layers, i.e., small round cysts approximately 60 μm in diameter and large oval cysts approximately 60 × 200 μm in diameter. The small cysts were scattered in the area adjacent to the ora serrata, while the large cysts were located more posteriorly (Figure 2a). In the small cysts, a mass of uniform small round cells was observed in the fluid-filled cavity (Figure 2b), while numerous disorganized and fragmented cells were observed in the cavity of the large cysts (Figure 2c).

Since the inner cell nuclei of the small cysts barely stained with hematoxylin occupied the majority of the area of the cell body, the small cysts stained with nestin appeared as blood vessels containing erythrocytes (Figure 2b). Furthermore, in a preliminary study, the outer layer of small and large cysts was stained with vWF, a marker of vascular endothelial cells [25] (Figure 9a,b). To rule out the possibility of blood vessels with erythrocytes, nuclear DNA was stained with DAPI, and our findings revealed that the inner cell nuclei of both the small and large cysts were intensely stained with DAPI, even though an atypical donut-shaped staining pattern of DNA was observed (Figure 3a–d). Hence, we concluded that the inner cell nuclei occupying most of the cell body were lightly and homogeneously stained with hematoxylin, and that a thin rim of the cytoplasm was stained with the intermediate filament nestin.

It has been reported that vWF is expressed in the trophoblasts derived from the trophectoderm [26,27,28]. We also stained the cystic lesions with CDX2, CK18, and YAP-1, markers of trophectoderm [29], and we found that all of those markers were expressed in the cyst wall. CK18 and YAP-1 also express in the vascular endothelial cells, however, CDX2 does not [30,31]. Based on gross morphology using a stereomicroscope and expression of CDX2 in the cyst wall, we excluded the possibility of the blood vessels.

Unlike the inner cell nuclei of the small cysts, the nuclei of neighboring retinal cells were strongly stained with hematoxylin in a coarse granular pattern characteristic of the interphase nucleus [32] (Figure 4a). These findings presumably indicate that the inner cell nuclei of the small cysts contain mostly euchromatin, and that the nuclei of the common retinal cells are predominantly comprised of heterochromatin. The difference of the hematoxylin stainability between euchromatin and heterochromatin can be explained via the following reasons. Transcriptionally active and loosely packed euchromatin (low DNA density) in the inner cell nuclei is weakly stained with hematoxylin, while transcriptionally silent tightly packed heterochromatin (high DNA density) in the retinal cell nuclei is densely stained with hematoxylin (Figure 5b) [32,33].

The inner cells of the small cysts in the peripheral retina exhibited the following morphological and histochemical features that are similar to pluripotent stem cells such as ES cells and induced pluripotent stem (iPS) cells: (1) a small round shape, (2) a high nuclear/cytoplasmic (N/C) ratio, and (3) a euchromatin-rich nucleus [34,35,36]. Furthermore, the small cysts formed structures closely resembling the preimplantation blastocyst [13], i.e., both the small cyst and the blastocyst were composed of three components: an inner cell mass comprised of extremely small cells, an outer wall consisting of a single layer of flattened cells, and a fluid-filled cavity [13,14]. The inner cell mass of the blastocyst attaches to the trophectoderm at the embryonic pole or future site of implantation [37]. Likewise, the inner cell cluster of the small cysts was almost always attached to the outer layer in the area located on the scleral side of the small cysts, revealing that, as with the blastocyst, the small cysts had a distinct polarity (Figure 2b).

Based on the above findings, we immunostained retinal sections of the enucleated canine eye for several transcription factors expressed in the blastocyst, i.e., inner cell mass markers (Oct4, Nanog, and Sox2) [38] and a trophectoderm marker (CDX2) [39]. Subsequently, we found that the small cysts were intensely stained with all of the above blastocyst markers, indicating that the small cysts showed a similar transcription factor expression profile compared to that of the blastocyst.

However, the intracellular distribution of these blastocyst markers in the small cysts was somewhat different from the natural blastocyst (Table 1). As for the blastocyst, Oct4, Nanog, and Sox2 are positively expressed in the inner cell nuclei and are negatively expressed in its cytoplasms [40]. On the other hand, these pluripotent transcription factors were more densely stained in the inner cell cytoplasms of the small cysts compared with in its nuclei, and were most prominently stained in the area where the cluster of the inner cells attached to the cyst wall (Figure 4a,b and Figure 5a). Moreover, although CDX2 is localized in the trophectoderm nuclei of the blastocyst [41], it was intensely expressed in the inner cell cytoplasms of the small cysts beside their outer layer cells, seemingly corresponding to the trophectoderm (Figure 4a,b and Figure 5a). These results likely indicated that the genetic or epigenetic features of the small cysts were, to a certain degree, different from those of the blastocyst. 

A potential link between embryogenesis and carcinogenesis has recently been pointed out [42]. Cancer stem cells are generally small and round with a high N/C ratio [43,44,45], resembling the inner cells of both small cysts and blastocysts. Cytoplasmic Oct4, Nanog, and Sox2 observed in the inner cells of the small cysts are reportedly expressed in cancer stem cells, zygotes, and blastomeres [38,39,46]. Thus, the inner cells of the small cysts seem to be formed through accumulation of epigenetic (and possibly genetic) changes, similar to cancer stem cells or embryonic cells at very early stages of development.

Next, we stained the retinal sections for the RPE markers CK18 and RPE65 [47,48]. Although the small and large cysts were scarcely stained with RPE65, the surrounding tissue of those cysts was intensely stained, especially on the vitreal side (Figure 6c). Intraretinal migration of RPE cells has reportedly been observed in various retinal diseases, including lattice degeneration, age-related macular degeneration, and Sanfilippo syndrome (associated with retinal cysts) [18,19,49]. The morphology of the RPE65-positive surrounding cells showed a mesenchymal and fibroblast-like appearance [50] (Figure 6c), and RPE65 is generally accepted as a specific marker of RPE cells [19,47]. Hence, the surrounding tissue of the cysts appeared to be derived from RPE cells that migrated after undergoing the epithelial–mesenchymal transition (EMT) [51,52].

CK18, a marker of RPE cells and trophectoderm [47,53], was negatively stained in the surrounding tissue, and was positively stained in the small cysts, including the outer layer cells (Figure 7a). During the EMT process, RPE cells reportedly lose epithelial markers like E-cadherin and CK18 [54] and acquire mesenchymal characteristics such as invasiveness and motility [52]. Hence, we posit that intraretinally migrated RPE cells have a mesenchymal phenotype associated with the down-regulation of CK18, resulting in negative staining for CK18 in the pericystic tissue. Meanwhile, CK18 expressed in the outer layer cells of the small cysts (Figure 7a) seemed to be a phenotypic marker of trophectoderm-like cells in the cyst wall [53].

Based on the findings above, a question arises regarding the expression of RPE65 in the surrounding tissue of the cysts. It has been reported that the expression of RPE markers including E-cadherin, CK18, and RPE65 is decreased after EMT [54]. Hence, it follows that transdifferentiated RPE cells surrounding the cysts will possibly lose the expression of RPE65. However, transdifferentiated RPE cells occasionally express RPE65 in several pathological conditions of the retina, including fibrotic scars of neovascular age-related macular degeneration [55] and the subretinal membrane of proliferative vitreoretinopathy [56]. The findings in our previous study also showed RPE65 expression in the peripheral areas of lattice degeneration [18], i.e., areas in which RPE-associated hyperplasia frequently occurs [57]. Thus, since the gene expression behavior of the RPE-specific marker RPE65 is seemingly different from that of epithelial markers in RPE cells, such as E-cadherin and CK18, RPE65-positive CK18-negative fibroblast-like cells surrounding the cysts might surely be migrated RPE cells. Pericystic RPE cells may provide a supporting microenvironment for the generation of blastocyst-like structures through cell-to-cell contact and paracrine effects [58].

As described previously, the fibroblast-like cells surrounding the cysts expressed pluripotent markers Oct4, Nanog, and Sox2, yet pigment cells were not observed in that region (Figure 4a,c and Figure 5a). Post injury, RPE cells reportedly lose their pigmentation and transdifferentiate into cells expressing Sox2 [59]. Transdifferentiated RPE cells also express Oct4 and Nanog in vitro [60]. Thus, the findings in our above-described experiments, as well as those of previous reports, appear to support our assumption that the pericystic fibroblast-like cells are RPE cells that migrated intraretinally post transdifferentiation [61].

Lastly, we stained retinal sections with YAP1, a trophectoderm marker that is also expressed in RPE cells during EMT [62]. The staining pattern of YAP1 in the surrounding tissue of the small cysts (Figure 8a) was quite similar to that of RPE65 (Figure 6c)’ i.e., both YAP1 and RPE65 were more intensely stained on the vitreal side of the surrounding tissue. Therefore, YAP1 observed in the surrounding tissue was probably expressed by intraretinally migrated RPE cells, as was the expression of RPE65. Meanwhile, although the outer layer cells of the small cysts were positively stained with YAP1, they were negatively stained with RPE65. Hence, we speculate that YAP1 observed in the outer wall of the small cysts was expressed as a marker of the outer layer of blastocyst cells, namely, the trophectoderm [63].

In contrast to the inner cells of the small cysts (Figure 2b), those of the large cysts were disorganized and fragmented (Figure 2c). In addition, the expression of key pluripotent marker Oct4 [64] in the inner cells of the large cysts was remarkably decreased compared to that of the small cysts. These findings appear to suggest that the inner cells of the large cysts lost pluripotency [65] and gradually became apoptotic, as shown by the fragmented cellular configuration. The deterioration of the inner cells of the large cysts can probably be attributed to the following reasons: (1) insufficient blood supply as opposed to the endometrium, (2) inadequate cytokine environment, and (3) detachment-induced cell death (anoikis) [66,67,68].

Of note, the ocular tissue of fish, including the retina, continues to grow throughout their lifespan [69]. Moreover, it is increasingly accepted that retinal stem cells located at the retinal margin of fish are active and continuously proliferative, thereby adding new cells to the periphery of the existing retina throughout life [69,70,71]. On the other hand, it has been emphasized that in mammals, retinal stem cells at the retinal margin are quiescent, and have little regenerative capacity [72]. However, and as stated above, dogs, including beagles, develop myopia associated with an elongation of the axial length even in adulthood [10]. This seemingly implies that mitotically active retinal stem cells still reside at the margin of the adult canine retina and divide to give rise to new retinal cells.

In this study, the small cysts were located more peripheral, while the large cysts were located more posteriorly (Figure 2a). Based on the above-described assumption that the newly generated retinal cells are added from the margin adjacent to the ciliary body, the large cysts seem to appear earlier than the small cysts to displace posteriorly during eye growth. For a long period of time after emergence as the small cysts, their size may increase and subsequently form the large cysts.

Reportedly, nestin is expressed in dividing immature cells, including Muller cells, in the developing retina [73]. Nestin immunoreactivity in Muller cells progressively attenuates in the laminated retina shortly after birth, and is barely detected [8], while dedifferentiated Muller cells re-express nestin after retinal traumatic injury and degeneration [74,75]. The findings in a study by Bahtia et al. [76] showed that intense nestin staining co-expressing with CRALBP (cellular retinaldehyde-binding protein), a marker of Muller cells, is expressed in the marginal retina of the adult human eye. Notably, the findings in a study by Martínez-Navarrete et al. [77] revealed that these nestin-positive Muller glia-like cells in the retinal margin of adult humans co-express the dividing cell marker PCNA (proliferating cell nuclear antigen). That finding correlates with the findings in a study by Gilyarov [78] showing that nestin is regarded as a reliable marker of proliferating neural stem/progenitor cells. Consequently, although the adult mammalian retina is generally considered to lack the potential to regenerate [72], nestin-expressing Muller cells seem to add new retinal cells at the retinal margin by acting as stem/progenitor cells, even in adults [77].

The findings in our previous study revealed that nestin-positive Muller cells and nestin-positive red/green cones are present in the foveal slope and foveola, respectively, of the monkey retina [79]. Nestin is thought to be expressed in neurogenetic regions in the adult brain, namely, the subventricular zone and hippocampal dentate gyrus [80]. Hence, we posit that the fovea of adult primates may constitute the stem/progenitor cell niche as well as the ciliary marginal zone of the retina. Nestin-positive Muller cells and nestin-positive red/green cones residing in adult primate fovea presumably contribute to homeostatic and injury-induced regeneration, including closure of a macular hole [79].

It is well known that in the retina, the foveola contains the highest density of cone photoreceptor cells [81], and that it is composed of exclusively red/green cones [82,83]. Moreover, Ahnelt [82] and Williams [83] reported that beside the foveola, the cone-rich rim (located at the retinal margin, with a width of about 1 mm) has a particularly high density of cones. Very intriguingly, the histological structures of the cone-rich rim closely resemble the foveola, as there is an almost exclusive presence of red/green cones and Muller cells, the thickness is thin compared to other parts of the retina, the region is non-laminated, and it is avascular [82,83,84,85].

However, the cone-rich rim and the foveola differ, and quite significantly, i.e., the photoreceptor outer segments in the cone-rich rim are tiny or absent [85], while the outer segments of central foveolar cones are approximately twice as long as those of parafoveal cones [86]. However, ocular coherence tomography (OCT) examination frequently reveals that disruption of the ellipsoid zone is restricted in the foveolar region in various macular diseases, such as a macular microhole, acute retinal pigment epitheliitis, and laser-induced maculopathy [87,88,89], and OCT images of these lesions are at least partly regarded as a loss of cone outer segments [90]. Since the cone outer segments in the foveal retina of the primate fetus are tiny or absent [91,92], we infer that the foveolar cones lacking outer segments are immature cells with characteristics of retinal stem/progenitor cells resulting from dedifferentiation. Moreover, it appears that regeneration of the ellipsoid zone [93,94] is caused by redifferentiation of the dedifferentiated foveolar red/green cones.

Retinoblastoma is currently defined as a primary malignant intraocular neoplasm that arises from immature retinal cells (retinoblasts) within the developing retina [95]. It should be noted that the findings in previous studies by Nork et al. [95], Bogenmann et al. [96], and Xu et al. [97,98] showed that more than 95% of retinoblastoma cells expressed markers of red/green cones and possess the characteristics of red/green cone precursors. Since it is widely accepted that cancer arises from tissue stem/progenitor cells [99,100], the histological findings of retinoblastoma presented by Xu et al. [97,98] may provide evidence supporting our assumption that red/green cones lacking outer segments in the retinal margin are potential candidates for retinal stem cells. Thus, it can be deduced that immature Muller cells or immature red/green cones are the origin of the blastocyst-like structures found in the peripheral region of the beagle retina.

In mammals, only fertilized eggs of more than 100 μm in diameter and early blastomeres are totipotent cells able to differentiate into a blastocyst, thus giving rise to both the embryo and placenta [17]. Accordingly, we speculate that blastocyst-like structures found in the canine peripheral retina are derived from early blastomere-like totipotent cells generated via the reprogramming of resident retinal stem cells. Nestin, which is negatively expressed in the natural blastocyst, was positively stained in the small cysts. Although it has generally been considered that epigenetic memory is erased during the reprogramming process (including demethylation of DNA), it occasionally remains after reprogramming [101]. Hence, we posit that nestin-positive retinal stem cells acquire totipotency through epigenetic reprogramming to differentiate into blastocyst-like structures, and still retain nestin expression due to neural lineage origin. As mentioned before, the axial length of the canine eye continuously increases, even in adulthood [10], seemingly indicating that retinal stem cells in the peripheral retina of the adult dogs are still mitotically active to generate new retina. In short, blastocyst-like structures in the peripheral retina of beagles may be derived from active stem cells that acquire totipotency through reprogramming.

As shown in the microscopic images of immunostaining against CK18 and RPE65 (Figure 6a,c), the inner cell nuclei of the small cysts occasionally contained multiple clustered or dispersed rod-like structures intensely stained with hematoxylin. Judging from the morphology and hematoxylin-stainability, these intranuclear structures seemed to be chromosomes in metaphase (mostly in lateral view) [102]. The immunostaining described above showed that most of the inner cells of the small cysts simultaneously contained rod-like intranuclear structures (Figure 6a,c), presumably because the inner cells underwent synchronous cell division. Early blastomeres tend to divide synchronously [103]. Thus, the nuclear states of the inner cells within the small cysts are possibly similar to those of the embryonal cells in very early stages of development.

In mammals, immature oocytes are arrested in the first meiotic prophase, with an enlarged nucleus, termed ‘germinal vesicle’ (GV), usually containing a large round nucleolus [104]. The tetraploid nuclei of GV stage oocytes are classified into two types according to the chromatin configuration, i.e., the SN (surrounded nucleolus) and the NSN (non-surrounded nucleolus) [105]. In SN-type oocytes, chromatin condenses around the nucleolus forming a heterochromatin ring, while in NSN-type oocytes, diffuse euchromatin spreads throughout the nucleoplasm except the nucleolus [21]. As mentioned above, the nuclei of the small and large cysts exhibited a peculiar donut-like staining pattern with DAPI (Figure 3b,d), which closely resembles those of NSN-type oocytes [106]. DAPI binds to the adenine-thymine rich regions of DNA [107], and the nucleolus is comprised mostly of proteins and ribosomal RNA containing uracil instead of thymine [108]. Thus, the nucleolus of the GV is negatively stained with DAPI, resulting in the formation of the donut-like staining pattern [109]. We speculate that with DAPI staining of the small and large cysts, the central round unstained area of the nucleus consists of a nucleolus, as does the GV.

Unlike DAPI, hematoxylin binds to RNA as well as DNA and consequently stains the nucleolus of the NSN-type GV, because positively charged hematoxylin binds to the negatively charged phosphate groups in nucleic acid [110]. The nuclei of the small and large cysts were homogenously stained with hematoxylin and did not show a donut-like staining pattern, presumably because the DAPI-unstained round area corresponded to the nucleolus containing RNA instead of DNA. Meanwhile, the NSN-type GV composed of euchromatin is lightly stained by hematoxylin, as dye stainability of loosely packed euchromatin is less than that of tightly packed heterochromatin, as mentioned above. The nuclei of the small and large cysts as well as the NSN-type GV were more intensely stained with DAPI compared with hematoxylin, probably due to it being a fluorescent dye (DAPI) with high sensitivity.

It should be noted that we undertook this research to investigate the pathogenesis of peripheral cystoid degeneration. However, and both unexpectedly and quite surprisingly, we found peculiar cystic lesions containing an inner cell mass in the peripheral retina of the beagles. Since those cystic lesions morphologically resembled blastocysts, we reconsidered the purpose of the research, and ultimately focused on the elucidation of our findings presumably related to stem cell biology and early embryonic development.

As a result, we obtained the following evidence supporting our assumption that the cystic lesions in the canine peripheral retina were blastocyst-like structures: (1) morphological similarity, i.e., the cystic lesions of the canine peripheral retina possessed multiple tiny inner cells with a high N/C ratio, an outer layer (single layer of flattened cells), and a fluid-filled cavity, corresponding to an inner cell mass, a trophectoderm, and a blastocoel of the blastocyst, respectively, (2) the expression of blastocyst markers, i.e., all of the blastocyst markers we examined were expressed in the small cysts, although the expression sites differed from blastocyst in some cases. The expression of Oct4, an inner cell mass marker, was not observed in the inner cells of the large cysts, probably due to cell degeneration, and (3) the presence of polarity, i.e., the inner cell mass of the canine blastocyst-like lesions were always attached to their outer layer on the scleral side, meaning that the blastocyst-like structure had distinct polarity. Similarly, the inner cell mass of the blastocyst was always attached to the trophectoderm at the embryonic pole (i.e., the implantation site).

## 4. Materials and Methods

### 4.1. Animals and Fundoscopic Examination of the Peripheral Retina of the Dogs

All experiments were conducted in accordance with the ARVO Statement for Use of Animals in Ophthalmic and Vision Research and were approved by the Ethics Committee of Osaka Medical and Pharmaceutical University, Takatsuki-City, Osaka, Japan (Approval No. 30099). This study involved ten eyes of five 3-year-old (young adult) beagle dogs purchased from Japan SLC, Inc. (Hamamatsu, Japan). All dogs were housed in an air-conditioned room at a temperature of approximately 23 °C and a humidity of 60% with a 12/12 h light/dark cycle. All animals were fed regular commercially available dog food and provided with constant free access to tap water. Anesthesia was induced via an intramuscular injection of medetomidine (0.02 mg/kg) and midazolam (0.3 mg/kg), followed 10 min later by an intravenous injection of pentobarbital sodium (10 mg/kg). The peripheral retina was examined using a binocular indirect ophthalmoscope. Of the five dogs examined, apparent peripheral retinal cysts were identified in both eyes of two dogs that underwent fundus photography using a wide-angle fundus camera (RetCam; Clarity Medical Systems, Inc., Pleasanton, CA, USA).

### 4.2. Preparation of Histological Sections of the Canine Peripheral Retina

For the preparation of the histological sections of the peripheral retina, the two dogs that underwent fundus photography were euthanized with a lethal dose of potassium chloride injected intracardially under general anesthesia, and both eyes of each dog were enucleated. In those four eyes, after retinal flat mount preparation, the cystic lesions in the peripheral retina were identified using a stereomicroscope, and specimens of the cystic lesions were then cut into blocks and processed into 5 μm thick paraffin sections.

### 4.3. Nestin Immunostaining

The retina sections were immunostained with a primary antibody against nestin, an intermediate filament protein known as a neural stem cell marker. After the antigen retrieval procedure was performed, the retina sections were incubated with rabbit anti-nestin monoclonal antibody (SP103, ab105389; Abcam Ltd., Tokyo, Japan) as the primary antibody at a 1:100 dilution overnight at 4 °C. After 3-Amino 9-ethylcarbazole (AEC) color development, hematoxylin was used for nuclear counterstaining (blue).

### 4.4. DAPI Nuclear Staining

Nuclear staining of the retina sections was performed using DAPI (blue emission; Dojindo Laboratories, Kumamoto, Japan) alone, followed by observation under a fluorescence microscope.

### 4.5. Oct4, Nanog, Sox2, and CDX2 Immunostaining

The retina sections were immunostained with antibodies against Oct4, Nanog, and Sox2, markers of pluripotent stem cells, and CDX2, a marker of trophectoderm, and the following primary antibodies were used: rabbit anti-Oct4 polyclonal antibody (ab18976; Abcam) at a 1:50 dilution; goat anti-Nanog polyclonal antibody (ab77095; Abcam) at a 1:200 dilution; rabbit anti-SOX2 polyclonal antibody (ab97959; Abcam) at a 1:50 dilution; and rabbit anti-CDX2 monoclonal antibody (EPR2764Y, ab76541; Abcam) at a 1:80 dilution. For all antibodies, the sections were incubated overnight at 4 °C. After AEC color development, counterstaining with hematoxylin was performed.

### 4.6. CK18 and RPE65 Immunostaining

Immunostaining of the retina sections was performed using antibodies against CK18 and RPE65, markers of RPE cells (CK18 also being a trophoblast marker). For CK18 staining, mouse anti-CK18 monoclonal antibody (C-04, ab668; Abcam) was used as the primary antibody at a 1:100 dilution after the antigen retrieval procedure. For RPE65 staining, mouse anti-RPE65 antibody (401.8B11.3D9, ab78036; Abcam) was used as the primary antibody at a 1:200 dilution without an antigen retrieval procedure. For both antibodies, the sections were incubated overnight at 4 °C, with AEC then used as a chromogenic substrate. Hematoxylin nuclear staining was used as a counterstain. A monkey model neural retina was used as a negative control.

### 4.7. YAP1 Immunostaining

Immunostaining of the retina sections was performed using an antibody against YAP1 (which is present inside the nucleus in the trophectoderm [58]). Rabbit anti-YAP1 polyclonal antibody (NB110-58358; Novus Biologicals, LLC, Centennial, CO, USA) was used as the primary antibody at a 1:100 dilution after the antigen retrieval procedure. The sections were incubated overnight at 4 °C, with AEC then used for color development. Hematoxylin counterstaining was performed.

### 4.8. vWF Immunostaining

Immunostaining was performed using an antibody against vWF, a marker of vascular endothelial cells. Rabbit anti-human von Willebrand factor antibody (A0082; Agilent Dako, Carpinteria, CA, USA) was used as the primary antibody and antigen activation was performed at a dilution factor of 1:200.

## 5. Conclusions

In summary, the findings in this study revealed that the small solitary cysts in the canine peripheral retina exhibited blastocyst-like structures. However, the intracellular distribution of pluripotent factors expressed in the small cysts was somewhat different from that in a natural blastocyst. The small cysts also expressed nestin, a neural (retinal) stem/progenitor cell marker, and the surrounding tissue expressed RPE65, a specific marker of RPE cells. Based on the above immunohistological findings, we hypothesize that retinal stem/progenitor cells acquire totipotency through the interaction with intraretinally migrated RPE cells, resulting in the formation of blastocyst-like structures. Recently, a number of articles have been published on the in vitro generation of blastocyst-like structures using cancer tissues, ovarian surface epithelial cells, and pluripotent stem cells [111,112,113,114]. However, to the best of our knowledge, this is the first report of the in vivo generation of blastocyst-like structures, and our findings shed new light on the mechanism of homeostatic and injury-induced regeneration of the adult mammalian retina.

## Figures and Tables

**Figure 1 ijms-25-06045-f001:**
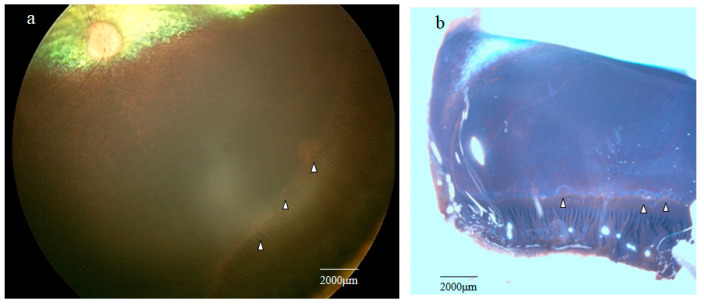
(**a**) RetCam image showing the presence of multiple, small, grayish cystic lesions (white arrowheads) in the most peripheral area of the retina adjacent to the ora serrata. (**b**) Stereoscopic microscopy image of an obtained peripheral retinal extension specimen showing cystic lesions (white arrowheads) at the most peripheral area of the retina near the ora serrata.

**Figure 2 ijms-25-06045-f002:**
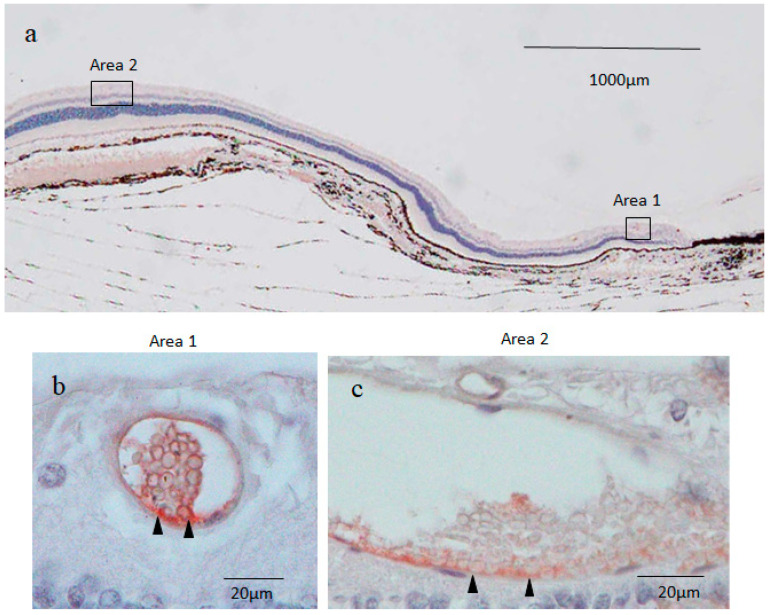
Immunostaining for nestin in the peripheral retina. (**a**) Low magnification images including the ora serrata showed that small cysts were scattered within the area 1000 μm from the ora serrata margin (area 1), with about two to three cysts per 1000 μm. On the other hand, large cysts were observed at a distance of more than 2000 μm from the ora serrata margin (area 2), with about two cysts per 1000 μm. More nestin-positive cells were observed in the small cysts than in the large cysts ((**b**,**c**); black arrowheads). In particular, strong staining was observed in the area where the cells in the cyst were in contact with the cyst wall. Hematoxylin was used for nuclear counterstaining.

**Figure 3 ijms-25-06045-f003:**
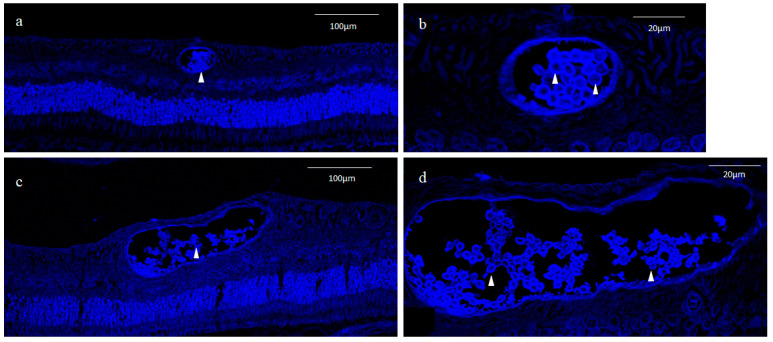
4′,6-diamidino-2-phenylindole (DAPI) staining. In the small cyst lumen, there was an accumulation of small round cells less than 10 μm in diameter with little cytoplasm, and DAPI staining showed that the nucleus occupied most of the inner part of each cell ((**a**,**b**); white arrowheads). In the large cysts, the cells in the cyst lumen were smaller than those in the small cyst lumen, and some of the cells were disintegrated and irregular in shape ((**c**,**d**); white arrowheads).

**Figure 4 ijms-25-06045-f004:**
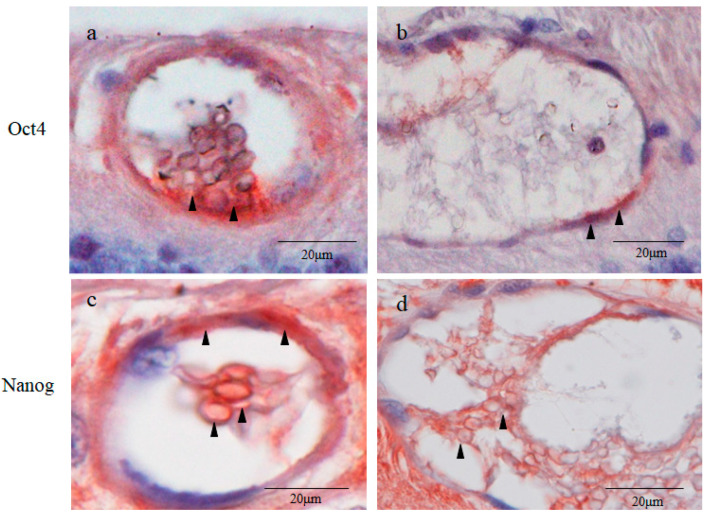
Immunostaining for octamer-binding transcription factor 4 (Oct4) and Nanog. Many Oct4-positive cells were found in the cell masses in the small cyst lumen and in the cyst wall, and strong staining was observed in the area where the cells in the cyst were in contact with the cyst wall ((**a**); black arrowheads). In the large cysts, only a few positive cells were found in the cyst wall ((**b**); black arrowheads). Nanog-positive cells were also abundant in the cell masses within the small cyst lumen and on the cyst wall surrounding the cyst ((**c**); black arrowheads). In the large cysts, Nanog-positive cells were also found in the cyst lumen, cyst wall, and surrounding area, but the staining was weaker than that in the small cysts ((**d**); black arrowheads).

**Figure 5 ijms-25-06045-f005:**
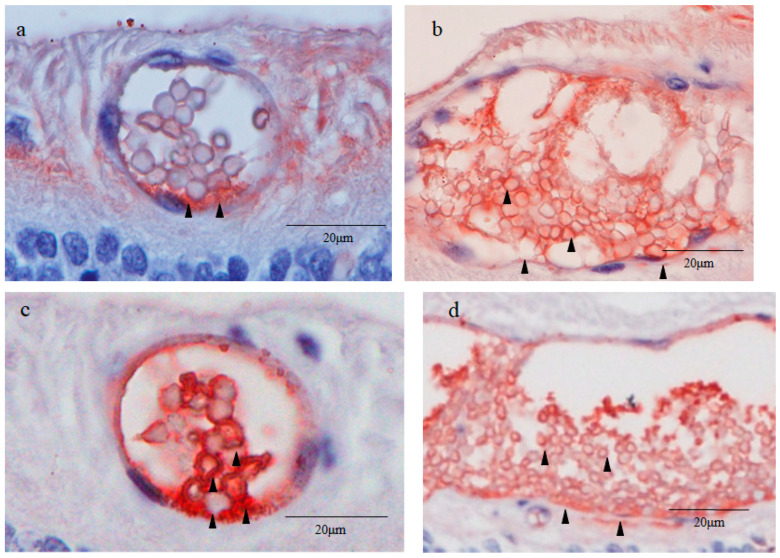
(**a**,**b**) Immunostaining for sex-determining region Y-box 2 (Sox2). Sox2-positive cells were found in the cell masses of the small cyst lumen and the cyst wall. As with Oct4, strong staining was observed in the area where the cells in the cyst were in contact with the cyst wall ((**a**), black arrowheads). In the large cysts, cells within the cyst and the cyst wall were partially positive, but the cell masses within the cyst stained more weakly than in small cysts ((**b**), black arrowheads). (**c**,**d**), Immunostaining for caudal type homeobox 2 (CDX2). CDX2-positive cells were found not only in the cyst wall but also the cell masses within the cyst lumen of both the small and large cysts, however, the staining was slightly stronger in the small cysts than in the large cysts ((**c**,**d**), black arrowheads).

**Figure 6 ijms-25-06045-f006:**
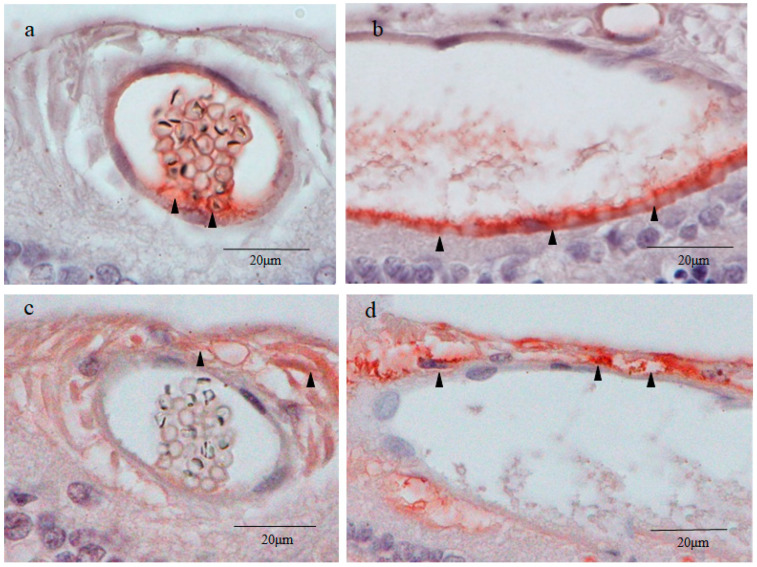
(**a**,**b**) Immunostaining for cytokeratin 18 (CK18) in small and large cysts. Strong immunostaining with CK18 was observed in both the inner cell mass and the cyst wall of the small cysts, and more intense staining was found in the area where the inner cell mass was in contact with the cyst wall ((**a**), black arrowheads). Rod-like structures were observed inside most of the inner cell nuclei in the small cysts. In the large cysts, cuboidal cells were continuous in a single layer on the retinal pigment epithelium side of the cyst wall, and strong staining was observed in that area, as well as weak staining in the cell mass within the cyst ((**b**), black arrowheads). (**c**,**d**) Immunostaining for retinal pigment epithelium 65 (RPE65). The intracystic cell mass and cyst wall of both small and large cysts showed weak staining for RPE65, and the staining was mainly observed in the vitreous side outside the cyst wall ((**c**,**d**), black arrowheads). Most of the inner cell nuclei in the small cysts contained rod-like structures, which may have been chromosomes. The inner cell nuclei in the large cysts did not show those findings.

**Figure 7 ijms-25-06045-f007:**
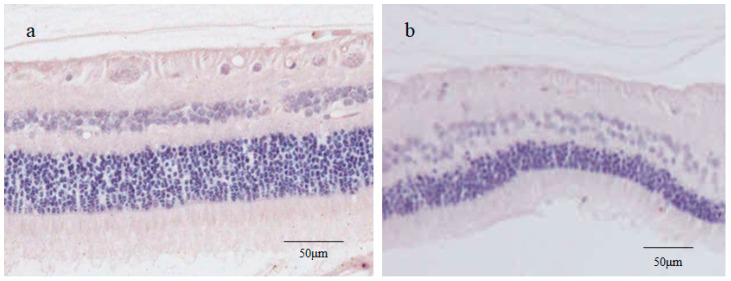
Immunostaining for CK18 (**a**) and RPE65 (**b**) in a monkey sensory retina. In monkey retinal sections used as control, there was no staining in the neural retina for either CK18 or RPE65.

**Figure 8 ijms-25-06045-f008:**
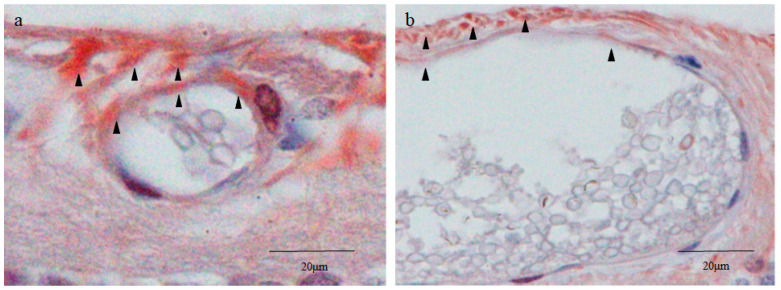
Immunostaining for yes-associated protein 1 (YAP1). In the small cysts, intense YAP1 staining was found on the vitreous side of the cyst wall and surrounding extracystic areas ((**a**), black arrowheads). In the large cysts, staining was found mainly on the vitreous side of the cyst wall and surrounding areas ((**b**), black arrowheads). The staining was similar to that of RPE65. There was no YAP1 staining in the intracystic cell mass (**a**,**b**).

**Figure 9 ijms-25-06045-f009:**
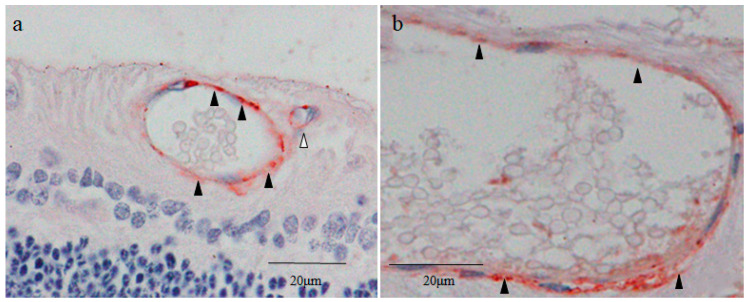
Immunostaining for von Willebrand factor (vWF). In the small cysts, vWF staining was found in the cyst wall ((**a**), black arrowheads) and vessel ((**a**), white arrowhead). In the large cysts, vWF staining was also found in the cyst wall ((**b**), black arrowheads). The white arrowhead in (**a**) indicates a lumen without inner cells (presumably a blood vessel).

**Table 1 ijms-25-06045-t001:** Immunostaining intensity of the markers of pluripotent cells and RPE cells in peripheral cysts and surrounding tissues.

	Small Cysts				Large Cysts			
Marker	Inner Cell Nuclei	Inner Cell Cytoplasms	Cyst Wall Cells	Surrounding Cells	Inner Cell Nuclei	Inner Cell Cytoplasms	Cyst Wall Cells	Surrounding Cells
Nestin	―	+++	+++	―	―	+	++	―
Oct4	++	+++	+++	++	―	―	++	+
Nanog	+	+++	+++	+++	―	++	+	++
Sox2	++	++	++	++	+	++	+	+
CDX2	++	+++	+++	―	+	++	++	―
CK18	+	+++	+++	―	―	+	+++	―
RPE65	―	―	―	+++	―	―	―	+++
YAP1	―	―	+++	+++	―	―	++	+++
vWF	―	―	+++	―	―	―	+++	―

+++ strongly positive, ++ moderately, + weakly positive, ― negative or scarcely positive.

## Data Availability

The data presented in this study are available on request from the corresponding author due to (specify the reason for the restriction).

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
