# Peer review of "Blastocyst-like Structures in the Peripheral Retina of Young Adult Beagles"

_ijms, 2024, doi:10.3390/ijms25116045_

Round 1
Reviewer 1 Report
Comments and Suggestions for Authors
The paper elucidated the molecular characteristics of the peripheral retinal cyst in 3 years old Beagles and identified blastocyst-like structures.
1. Fundus photos provided were not in high quality. The cyst seems to locate at where the retina stopping to develop, ie., a pathology similar to retinopathy of prematurity. Whether this cyst like structure is dedicated to studying peripheral retinal degeneration is questionable.
2. It is rarely reported of the cyst like structures in the peripheral retina in other species. The main findings of this study need to be verified.
3. These cell wall of the cyst need to be verified if they are endothelial cells.
4. The cells within the cysts were composed with pluripotent stem like characters, however; many of them were not double positive with DAPI.
Comments on the Quality of English Language
No concerns for me.
Reviewer 2 Report
Comments and Suggestions for Authors
The manuscript essentially describes a totally novel biological
phenomenon related to blastocysts. The authors maintain focus on this
throughout the manuscript. The authors have identified small cysts in
the retina that contain both stem cells and retinal pigment epithelial
cells. These structures have features of blastocysts. Taking into
account the key role of blastocyst for providing embryonic stem cells,
the study is significant and important for large bio-community. The
outcome as well as the appraoch is very unique. The discussion is an
absolute joy to read. It is very well written. The text is clear and
easy to read and this is one of the strengths of the manuscript. This is
a very interesting study that has been carefully conducted with valid
scientific methods. I have no further comments.
Reviewer 3 Report
Comments and Suggestions for Authors
Blastocyst-Like Structures in the Peripheral Retina of Young 2
Adult Beagles.
Peripheral retinal degenerations include typical peripheral cystoid and reticular degeneration , network degeneration , and cobblestone degeneration. Typical peripheral cystoid degeneration, consisting of a multitude of compact microcysts, is considered to be age-dependent because its prevalence increases with age . However, solitary peripheral cysts have occasionally been observed in younger generations.
Blastocysts are composed of an inner cell mass (embryoblast), an outer cell layer (trophectoderm), and a fluid-filled cavity (blastocele). The internal cells of a blastocyst, characterized by large nuclei and scant cytoplasm, have the ability to differentiate into all somatic cells of the body and can be isolated and cultured to produce pluripotent embryonic stem (ES) cells. It has been reported that the trophectoderm gives rise to placental trophoblasts and totipotent cells, such as fer-.
tilized eggs and blastomeres (probably up to the four-cell stage), have the ability to form blastocysts that differentiate into extraembryonic tissues (i.e. placenta, yolk sac, etc.) In addition to all somatic cells .
The topic presented by the authors is very interesting and very worrying for new therapies for wet and dry AMD, as well as analyzing blastocyst-like structures in the peripheral retina.
After reading the entire manuscript, I have observed that the literature consulted for this manuscript only has 109 citations, although the authors could have further enriched the manuscript with new and more up-to-date citations on AMD.
In relation to these observations, I have a few questions:
1. Can the authors explain the biochemical mechanism of blastocysts in the peripheral retina?
2- What importance does or does the immunohistological study of peripheral retinal cysts in AMD have on the regulation of angiogenesis?
3. How are sex-determining region Y-box 2 (Sox2) and caudal-type homeobox 2 (CDX2) immunostaining involved?
3. Could you explain the pharmacological mechanism of neovascularization to treat retinal cysts?
